# Toxicity, Sublethal and Low Dose Effects of Imidacloprid and Deltamethrin on the Aphidophagous Predator *Ceratomegilla undecimnotata* (Coleoptera: Coccinellidae)

**DOI:** 10.3390/insects12080696

**Published:** 2021-08-03

**Authors:** Panagiotis J. Skouras, Anastasios I. Darras, Marina Mprokaki, Vasilios Demopoulos, John T. Margaritopoulos, Costas Delis, George J. Stathas

**Affiliations:** 1Laboratory of Agricultural Entomology and Zoology, Department of Agriculture, Kalamata Campus, University of the Peloponnese, 24100 Antikalamos, Greece; marina.brokaki.97@gmail.com (M.M.); g.stathas@uop.gr (G.J.S.); 2Department of Agriculture, Kalamata Campus, University of the Peloponnese, 24100 Antikalamos, Greece; a.darras@uop.gr (A.I.D.); delis@us.uop.gr (C.D.); 3Laboratory of Plant Protection, Department of Agriculture, Kalamata Campus, University of the Peloponnese, 24100 Antikalamos, Greece; v.dimopoulos@go.uop.gr; 4Department of Plant Protection, Institute of Industrial and Fodder Crops, Hellenic Agricultural Organization “DEMETER”—NAGREF, 38446 Volos, Greece; johnmargaritopoulos@gmail.com

**Keywords:** biological control, biological control agents, ecotoxicology, integrated pest management, neonicotinoid, pesticide toxicity, pyrethroids, side effects

## Abstract

**Simple Summary:**

Chemical insecticides are used to control agricultural pests all over the world. However, extensive use of chemical insecticides can be harmful to human health and negatively impact the environment and biological control agents. We studied the toxicity and sublethal effects of imidacloprid and deltamethrin on the aphidophagous coccinellid predator *Ceratomegilla undecimnotata* (Coleoptera: Coccinellidae). We evaluated one low (LD_30_) and one sublethal dose (LD_10_) for both insecticides. Lethal and sublethal effects of both insecticides negatively affected survival, development, reproduction, and longevity, and reduced the intrinsic (*r*) and finite (*λ*) rate of increase and the net reproduction rate (*R_0_*) of treated populations compared to the control. Our findings indicate that the use of imidacloprid and deltamethrin in combination with *C. undecimnotata* in the context of IPM should be taken with caution due to the toxic effects of the chemicals in the biocontrol agent under laboratory conditions.

**Abstract:**

*Ceratomegilla undecimnotata* (Coleoptera: Coccinellidae) is a common aphidophagous coccinellid predator used in biological control against aphid pests. Knowing toxicity, lethal, and sublethal effects of insecticides on natural enemies is essential in order to incorporate them into Integrated Pest Management (IPM). In the present study, the lethal and sublethal effects of imidacloprid and deltamethrin were evaluated on the fourth instar larvae of *C. undecimnotata* and subsequently on the full life cycle. Our results strongly suggest that sublethal and low doses of imidacloprid and deltamethrin at LD_10_ and LD_30_ affected fourth instar larvae duration time, adult preoviposition period, total preoviposition period, and fecundity. Moreover, the intrinsic (*r*) and finite (*λ*) rate of increase and the net reproduction rate (R_0_) significantly decreased in populations treated with imidacloprid compared to the control population. The data clearly suggest that imidacloprid and deltamethrin have a negative influence on population growth parameters of *C. undecimnotata* at sublethal and low doses and, therefore, these insecticides should be used with caution within the context of IPM.

## 1. Introduction

Coccinellids are among the most important predators against many pests including aphids, coccids and mites [1]. The predatory ladybird beetle *Ceratomegilla*
*undecimnotata* (Schneider) (Coleoptera: Coccinellidae) is one of the most common ladybird species in Greece [2,3] and preys only on a few aphid species, such as *Myzus persicae* (Sulzer) and *Aphis fabae* Scopoli (Hemiptera: Aphididae) [4]. *Ceratomegilla undecimnotata* have been reported on peach orchards, tobacco, cotton, and maize, suppressing aphid populations [2,3,5]. The larvae and adults of *C. undecimnotata* prey on aphids, exhibiting high voracity and specificity, and a high population rate increase [4,6].

Neonicotinoids, especially imidacloprid, thiamethoxam, and clothianidin, have a market share greater than 20% of total global insecticide sales due to their high efficacy against a large number of pests and their versatility of use [7]. Imidacloprid is a broad-spectrum neonicotinoid insecticide acting against piercing/sucking insects, such as aphids, whiteflies, leafhoppers, and various coleopteran pests. It can be applied in a variety of ways, such as sprays, root drenches, and seed dressing [7]. Imidacloprid acts as a competitive agonist of the insect nicotinic acetylcholine receptor (nAChR) in post-synaptic membranes (IRAC MoA group 4) [7,8]. On the other hand, as a common alternative to neonicotinoids, pyrethroids act by binding to the voltage gate sodium channel and is used against pests like Diptera and Hemiptera, especially Aphids, Lepidoptera and Coleoptera [9,10]. Deltamethrin is a type II pyrethroid insecticide used against many agricultural and non-agricultural pest [9].

The widespread use of neonicotinoids, due to their high effectiveness against a range of pests, causes a number of serious issues including food contamination, environmental pollution, the development of resistance in many pests, secondary pest outbreaks, and effects on populations of natural enemies, and they have a negative impact on behavioral and/or physiological traits [7,10,11,12,13,14,15,16,17,18,19,20,21]. Recently, three neonicotinoid insecticides (imidacloprid, thiamethoxam, and clothianidin) were banned from use in the EU on all field crops [22]; however, their application in protected crops is still authorized [13]. On the other hand, neonicotinoids are still widely used in other countries such as the USA and China [23]. In all cases, it is essential to develop Integrated Pest Management (IPM) programs that can control pest populations below the economic threshold but without relying solely on chemical compounds. In other words, IPM strategies should combine the use of selective insecticides and natural enemies [24,25]. Therefore, before the implementation of natural enemies in an IPM program, it is substantial to evaluate the toxicity and sublethal effects of the insecticides [11,13].

Coccinellid predators, such as *C. undecimnotata*, may be directly exposed to spray droplets of insecticides [24], contact residues on foliage [26], or feed with insecticide-contaminated pests [27]. Besides mortality (a lethal effect), insecticides may cause sublethal effects on biological and behavioral parameters including developmental time, fecundity, longevity, sex ratio, feeding activity, predation rate, orientation, and mobility [12,28], thus minimizing the potential of the biological control agents. Previous studies have reported the impact of imidacloprid on Coccinellid predators [14,29,30,31,32]. Imidacloprid has been reported to reduce population growth parameters in *Coccinella septempunctata* [29], *Hippodamia convergens* [30], and *Hippodamia variegata* (Colleoptera: Coccinellidae) [14]. Imidacloprid has been found to reduce fecundity in coccinellids such as *H. variegata* and *Nephaspis oculata* [14,32] and affected predators’ functional responses in *Serangium japonicum* and *Harmonia axyridis* (Colleoptera: Coccinellidae) [33,34]. On the other hand, only limited data are available on the impact of deltamethrin on Coccinellid predators [35].

Life table analysis is a research tool used to analyze and understand the effect of pesticides on demographic parameters and survival rates. The life table analysis has been adapted by many authors during the last few decades due to the advantages considering age, stage, and sex differences [36]. Female age-specific life tables do not take into consideration both the stage and male population differentiation and results in incorrect data analysis and/or interpretation [37].

No previous data were found on the toxicity and sublethal effects of imidacloprid on *C. undecimnotata.* The present study investigated the toxicity, sublethal, and low dose effects of imidacloprid and deltamethrin on *C. undecimnotata* including demographic parameters and survival rates. The results could be relevant for the development of IPM programs and could help optimize the use of two common insecticides along with *C. undecimnotata* against *A. fabae* and *M. persicae* in field and greenhouse crops.

## 2. Materials and Methods

### 2.1. Insecticide Properties and Treatments

In the experiments, two commercial insecticides were used: the 20% soluble liquid imidacloprid-based commercial product Confidor Forte 200SL, (Bayer Crop Science, Athens, Greece) and the 2.5% emulsifiable concentrate deltamethrin (Decis 25EC, Bayer Crop Science, Greece). Both insecticides were diluted with HPLC-grade acetone (Merk, Darmstadt, Germany). Imidacloprid was dissolved in acetone to produce a dose series of 320, 160, 80, 40, 20, 10, 5, ng a.i./insect in order to determine the sublethal and low dose (Appendix A). The corresponding deltamethrin doses were 35, 17.5, 8.75, 4.38, 2.19, 1.09, 0.55, 0.27, and 0.14 ng a.i./insect.

### 2.2. Insects and Plant Material

The *C. undecimnotata* colony was initiated from individuals (100 adults) collected in 2017 from tobacco fields in the area of Meliki (northern Greece). *A. fabae* were collected on broad bean plants leaves and stems (*Vicia faba* L. (Fabaceae)) in Kalamata, Messinia, Southern Greece. Both were transported to the Laboratory of Agricultural Entomology and Zoology of the Department of Agricultural Science at the University of the Peloponnese (Kalamata, Prefecture of Messinia, Greece). Laboratory rearing of *C. undecimnotata* took place in cylindrical acrylic glass containers (30 cm diameter × 50 cm height.) kept in a controlled environment chamber (Sanyo MLR-351H) (25 ± 1 °C, 65 ± 2% RH and 16L:8D photoperiod). Coccinellids were reared on broad beans infested with a mixed instar of broad bean aphid, *A. fabae*. The *A. fabae* colony was maintained on *V. faba*. The plants were kept in cages (40 × 45 × 50 cm) with a wooden washable flush floor; the sides and the top were covered with a fine aphid-proof muslin to allow aeration and prevent aphid escape. The cages were kept inside a controlled environment chamber at 20 ± 1 °C, 50% ± 5% RH and 16L:8D photoperiod.

### 2.3. Lethal Toxicity

Lethal toxicity of imidacloprid and deltamethrin was assessed on fourth instar larvae of the predator *C. undecimnotata.* Eggs (<24 h old) of the F1 generation were randomly selected from the *C. undecimnotata* laboratory colony and were transferred to a 9-cm-diameter Petri dish. The Petri dishes were then placed inside a growth chamber. After egg hatching, each larva was transferred individually into a Blackman box (7.7 × 4.5 × 2.0 cm) (Blackman 1971) and was fed *A. fabae* ad libitum. The toxicity of the imidacloprid doses to the fourth instar larvae (12–24 h old) of *C. undecimnotata* was assessed, using a Hamilton microsyringe. Insecticide was applied in 1 μL of acetone on the mesonotum of each larva using a 10 μL Hamilton microsyringe. Controls were treated with acetone alone. Four replications of ten fourth instar larvae were used per dose. A preliminary experiment was conducted to determine the range of doses by exposing the predator to imidacloprid (6, 60, and 300 ng imidacloprid/insect) or deltamethrin (1.75, 17.5, and 87.5 ng deltamethrin/insect) at decreasing doses until mortality rates below 100% were observed. The treated and control insects were placed in Blackman boxes and kept inside an environmental growth chamber at standard environmental conditions, (25 ± 1 °C, 65% ± 5% RH and 16L:8D photoperiod). Larvae of *C. undecimnotata* were fed every day with mixed live aphid instars ad libitum. Mortality was recorded three days after exposure. Treated larvae were scored dead if they did not react when pushed with a brush.

### 2.4. Life History Study

Impact of Insecticides on the Development, Survival, Reproduction, and Population Parameters of *C*. *undecimnotata*

The sublethal and low dose effects of imidacloprid and deltamethrin at LD_10_ and LD_30_ doses were previously calculated from the lethal toxicity bioassay. Three hundred and eight eggs (<24 h old) of the F1 generation of *C. undecimnotata* were randomly selected to construct the corresponding life table according to the description of Nawaz [38]. A total of 40 eggs for the control group and 63 and 80 eggs in the LD_10_ and LD_30_ of imidacloprid, and 45 and 80 eggs in the LD_10_ and LD_30_ of deltamethrin treatment groups, were used, respectively. Newly laid eggs of *C. undecimnotata* were transferred in petri dishes and incubation period and hatch rate of eggs was recorded daily. After egg hatching, each larva was transferred individually into a Blackman box, at the base of which there was a piece of water-saturated moss [39]. One hundred and fifty young adults of *A. fabae* were placed inside the Blackman box every 24 h [4]. The larvae were examined once every day, and mortality and molting were registered. Two doses (i.e., 10.68 and 20.95 mg a.i. L^−1^) of imidacloprid and two doses (i.e., 0.77 and 1.76 mg a.i. L^−1^) of deltamethrin equivalent to LD_10_ and LD_30_ doses, respectively, were prepared and fourth instar larvae (<12 h old) were treated. Each egg was considered as one replicate for each treatment. Egg hatchability, egg, larvae, and pupae duration times, and molting were recorded daily until the molt of adults. When the adults emerged from each treatment (<24 h old), female and male individuals were paired and transferred with a paintbrush to a new Blackman box. Mortality, male and female longevity, oviposition period, adult preoviposition period, total preoviposition period, and fecundity were recorded daily until the death of both female and male predators. Approximately 200 young adults of *A. fabae* were placed into the Blackman box every 24 h. Fourth instar larvae and rearing conditions were chosen because when first and second instar larvae were tested, the natural mortality was high and fourth instar larval had a higher voracity compared with other larvae instars [4]. The treated and control insects were kept in Blackman boxes and maintained in an environmental growth chamber at standard environmental conditions (25 ± 1 °C, 65 ± 5% RH and 16L:8D photoperiod).

### 2.5. Data Analysis

The dose–mortality relationship was determined by probit analysis using SPSS version 18.0 (SPSS Inc., Chicago, IL, USA) to determine the LD_10_ and LD_30_ values with 95% confidence intervals. The effects of sublethal and low doses on fourth instar larvae and pupae developmental times, adult preoviposition period (APOP), total preoviposition period (TPOP), longevity, and the fecundity parameters in each treatment were analyzed via the age-stage, two-sex life table theory [40,41]. The basic life table parameters, where *x* denotes age and *j* denotes stage, such as age-specific survival rate (*l_x_*), age-stage specific survival rate (*s_xj_*), age-stage specific fecundity (*f_xj_*), age-specific fecundity (*m_x_*), age-specific maternity (*l_x_m_x_*), age-stage specific reproductivity (*v_xj_*), life expectancy (*e_xj_*) and population parameters were calculated using the computer program TWOSEX-MS Chart [42].

The net reproductive rate (*R*_0_) is calculated as: $R_{0}=\overset{\infty }{\underset{x=0}{\sum }}\;l{\;}_{x}m_{x}$.

Intrinsic rate of increase (r): $\sum_{x=0}^{\infty }e^{-r\left(x+1\right)}l_{x}m_{x}=1$ [43,44,45].

Mean generation time (*T*): $T=\frac{\ln R_{0}}{r}$ [46].

Finite rate of increase (λ): *λ* = $e^{r}$ [46].

The variances and standard errors of population parameters were estimated by using a bootstrap technique with 100,000 random resamples [47,48] and the paired bootstrap test was used for the comparison of treatments [37,49]. Both techniques were included in TWOSEX-MS Chart [42]. SigmaPlot 12.0 software (Systat Software Inc., San Jose, CA, USA) was used to create curves for life expectancy, fecundity, survival rate, reproductive values, and population projection.

## 3. Results

### 3.1. Toxicity of Imidacloprid and Deltamethrin on Fourth Instar Larvae of C. undecimnotata

Results on the lethal and sublethal toxicity of imidacloprid and deltamethrin to *C. undecimnotata* are reported in Table 1, Appendix A. The results showed that the LD_10_ and LD_30_ of imidacloprid to *C. undecimnotata* were 10.68 and 20.95 ng a.i. per insect, respectively. The corresponding values for deltamethrin were 0.77 and 1.76 ng deltamethrin per insect, respectively. No mortality was recorded in the control group. The LD_10_ were used as the sublethal dose (X^2^ = 2.13, df = 1, *p* = 0.145, for imidacloprid and X^2^ = 3.40, df = 1, *p* = 0.065, for deltamethrin) and low dose (*p* < 0.05) for the life tables for both insecticides, respectively.

### 3.2. Effects of Imidacloprid and Deltamethrin on the Developmental Duration Time, Longevity and Fecundity of C. undecimnotata

Table 2 shows the effect of imidacloprid and deltamethrin on the developmental duration time of *C. undecimnotata* fourth instar larvae and pupae. The results demonstrate that the development time of fourth instar larvae was significantly (*p* < 0.01) prolonged in the imidacloprid groups (LD_10_ = 5.22 days and LD_30_ = 5.31 days) compared to the control (4.06 days). The developmental duration of fourth instar larvae was significantly longer (*p* = 0.09 for LD_10_ and *p* = 0.001 for LD_30_) for deltamethrin compared to the control group. No significant differences between control and imidacloprid or deltamethrin groups (*p* < 0.05) were observed in the development time of pupae.

The mean fecundity of the control group was significantly higher (773.34 eggs) than in the LD_30_ treatment of imidacloprid (*p* = 0.014) (476.6 eggs) (Table 3). The APOP and TPOP were extended significantly (*p* = 0.01 for APOP and *p* = 0.001 for TPOP) by the low dose of imidacloprid LD_30_ (9.00 and 28.27 days, respectively) compared to the control group (4.76 and 22.18 days, respectively). No significant difference was found (*p* > 0.05) between control and deltamethrin doses. The female adult longevity was reduced significantly (*p* = 0.049) by the low dose of imidacloprid LD_30_ compared to the control group. Furthermore, there were significant (*p* = 0.034) differences between the sublethal dose of deltamethrin LD_10_ and the control group in terms of male adult longevity (Table 3).

### 3.3. Influence of Imidacloprid and Deltamethrin on the Population Parameters of C. undecimnotata

The results show that there was a significantly negative effect of the low dose of imidacloprid and deltamethrin on the population parameters, such as intrinsic (*r*) and finite rate of increase (*λ*), in comparison to the control (Table 4). Mean generation time (*T*) and net reproductive rate (*R_0_*) were significantly different between imidacloprid (both doses) and control.

The age-stage specific survival rates *s_xj_* curve, which represents the probability that a newborn will survive to age *x* and stage *j,* is shown in Figure 1. With increasing imidacloprid and deltamethrin dose, the probability of survival from larva to pupa and adulthood decreased after exposure to the two sublethal treatments, respectively. The peak survival rates for females and males in the control were 42.5% and 40.0%, respectively, and lasted 32 and 13 days, respectively. In contrast, the peak survival rate value in the LD_10_ and LD_30_ treatment groups decreased proportionally as imidacloprid and deltamethrin doses increased (i.e., imidacloprid: LD_10_: 25.4% for females (last 35 days) and 31.8% for males (last 3 days) and LD_30_: 18.8% for females (last 43 days), and 21.3% for males (last 22 days); deltamethrin: LD_10_: 31.1% for females (last 35 days) and 35.6% for males (last 21 days) and LD_30_: 18.75% for females (last 16 days), and 21.3% for males (last 14 days)).

Age-specific survival rate *l_x_*, fecundity of the total population *m_x_*, and net maternity *l_x_m_x_* are shown in Figure 2. The age-specific survival rate *l_x_* shows the probability of newborn eggs surviving to age *x*, irrespective of the stage differentiation. The *l_x_* decreased sharply when the fourth instar larvae were treated to sublethal and low doses of imidacloprid and deltamethrin. The highest *m_x_* (11.30 eggs individual^−1^ days^−1^) occurred on the 29th day for the control. However, the highest *m_x_* for imidacloprid (9.88 and 7.93 eggs individual^−1^ days^−1^ for LD_10_ and LD_30_, respectively) occurred on the 97th and 57th day of the LD_10_ and LD_30_ treatments, respectively. The highest *m_x_* for deltamethrin (13.00 and 10.33 eggs individual^−1^ days^−1^ for LD_10_ and LD_30_, respectively) occurred on the 127th and 149th day of the LD_10_ and LD_30_ treatments, respectively (Figure 2). In general, the values of age-specific maternity (*l_x_m_x_*) of *C. undecimnotata* were higher in the control than in imidacloprid and deltamethrin treatments.

The reproductive value (*v_xj_*) curves are shown in Figure 3. There were no effects on the reproductive values of larvae when fourth instar larvae of *C. undecimnotata* were exposed to imidacloprid. The peak reproductive values of fourth instar larvae under the LD_30_ treatment with deltamethrin were higher than that of the control, while under the LD_10_ treatment, the peak was lower than that of the control insects. Imidacloprid decreased the female reproductive values. The highest peak (114.27 day^−1^) for the controls occurred on the 29th day. However, the highest peaks for imidacloprid on the LD_10_ and LD_30_ treatments were 93.62 day^−1^ (on the 33rd day) and 113.86 day^−1^ (on the 48th day), respectively. In the case of deltamethrin, the corresponding values of LD_10_ and LD_30_ treatments were 102.56 day^−1^ (on the 42nd day) and 120.81 day^−1^ (on the 38th day), respectively.

The life expectancy curves (*e_xj_*) for each age-stage group of *C. undecimnotata* are presented in Figure 4. Imidacloprid significantly decreased the *e_xj_* of the egg, larval, pupal, female, and male stages at LD_10_ and LD_30_, compared to the control group. The *e_xj_* values for newly hatched *C. undecimnotata* eggs of the imidacloprid groups were 51.7 and 37.8 days in the LD_10_ and LD_30_ groups, respectively. The corresponding values for deltamethrin were 59.5 and 38.4 in the LD_10_ and LD_30_ groups, respectively, while in the control it was 83.1 days.

The population growth of *C. undecimnotata* projected by using the life table data is shown in Figure 5. The size of the *C. undecimnotata* population was larger and the total population growth was faster in the control group than the imidacloprid and deltamethrin groups.

## 4. Discussion

In this study, we provided experimental evidence of the toxicity and the sublethal and low dose effect of imidacloprid and deltamethrin on the predator *C. undecimnotata.* Imidacloprid and deltamethrin doses increased fourth instar larvae development time and decreased various life parameters. The two sublethal doses of both insecticides tested negatively affected the population growth parameters of the predator.

The LD_50_ values of imidacloprid and deltamethrin to the *C. undecimnotata* fourth instar larvae were half and one fifth the dose recommended to control aphids in crops, respectively [35]. Similarly, the insecticide imidacloprid is highly toxic to other coccinellid predators, such us *Hippodamia variegata* (Goeze) (Coleoptera: Coccinellidae) [14,31], and third instar larvae of *Coleomegilla maculata* (De Geer) (Coleoptera: Coccinellidae) [50]. Deltamethrin was the most toxic and similar toxic effects by other insecticides with the same mode of action have been observed on many coccinellids such us *Adalia bipunctata* (L.) (Coleoptera: Coccinellidae) [51] and *C. septempunctata* [35]. In general, chemical insecticides are more harmful to first, second, and third than to fourth instar larvae [21,31,52]. Yao et al. [52] reported that the lowest LD_50_ for the neonicotinoid thiamethoxam was associated with its systemic application, while contact of the target with dried residue produced the highest value. Moreover, predator mortality due to pesticides can vary with exposure types, such as contact with exudates and plant tissues and/or feeding with treated pests and honeydew [22]. Jan and Moosa [53] reported that insecticide application in the laboratory may overestimate the toxicity, which might be different to that recorded on field trials.

Apart from acute toxicity, insecticide exposure may induce stress and lead to various secondary effects. According to our results, exposure of fourth instar larvae to sublethal doses of imidacloprid and deltamethrin (LD_10_ and LD_30_) extended the fourth instar larvae duration time compared to the control group. Comparing our results with similar studies, the sublethal and low doses of imidacloprid extended the larvae development time in the coccinellid predators *H. variegata* [14,31], *C. septempunctata* [29,31,35], and *Harmonia axyridis* (Coleoptera: Coccinellidae) [54]. The prolonged developmental time of treated fourth instar larvae is a result of the reduction of larval fitness due to lower food intake [14]. Imidacloprid has also been found to disrupt the hormonal balance, negatively affect nutritional absorption, and/or stimulate a natural adaptation of predator reproductive strategies to reduced feeding [55]. At the same time, the predators redirect energy towards detoxifying the insecticide instead of growth and development [56]. Further research is required in pesticide risk-assessment evaluation schemes to determine the sublethal effects on predatory behavior (e.g., functional response) [24,25,33,57].

Besides prolonged larval development time, imidacloprid reduced the fecundity, female oviposition period, and adult longevity of the predator. The sublethal doses LD_10_ and LD_30_ reduced fecundity by 23.5% and 38.4%, respectively, compared to the control group. A decrease in the number of eggs laid by females was also found in other coccinellid species, such us *C. septempunctata* [29,58], *H. variegata* [14], *Rodolia cardinalis* (Coleoptera: Coccinellidae) [59], and *Hippodamia undecimnotata* (Coleoptera: Coccinellidae) [60]. The results indicate that female fecundity may be decreased due to the reduction of the oviposition period or the negative effect of imidacloprid on the feeding efficiency of *C. undecimnotata*. The latter was found in the study of Xie et al. [61] on the coccinellid predator *Cryptolaemus montrouzieri* (Coleoptera: Coccinellidae).

Our results confirm that sublethal doses of imidacloprid impaired the population parameters *r*, *λ*, *Τ*, and *R_0_*. The values of those parameters suggested that the low doses of imidacloprid and deltamethrin decreased population growth of *C. undecimnotata*. Overuse of imidacloprid and deltamethrin in agricultural ecosystems may reduce population increases and, as a result, the effectiveness of *C. undecimnotata.* Imidacloprid prolonged the APOP and TPOP in LD_10_ and LD_30_ treatment and reduced the intrinsic rate of increase (*r*); this was the lowest when we compared LD_10_ and LD_30_ to the control treatments. These findings are in line with the results reported by other authors [62], who found that sublethal doses of sulfoxaflor increased TPOP and were negatively correlated with *r* of *Sogatella furcifera* (Hemiptera: Delphacidae). The reduction of population parameters *r*, *λ*, and *R_0_* of imidacloprid- and deltamethrin-treated populations compared to the control shows that low doses of both insecticides may have a long-term negative effect on insect physiology [38].

The *s_xj_* value was also affected; this measure of survival rate decreased in accordance with the increase in imidacloprid and the dose to fourth instar larvae, pupae and adults. In addition, *C. undecimnotata* also demonstrated different population *l_x_*, *m_x_*, *l_x_m_x_,* and *v_xj_* values when treated with different imidacloprid or deltamethrin doses. The parameters *m_x_*, *l_x_m_x_,* and *v_xj_* were measurably higher in the control population, which indicates that sublethal or low doses of imidacloprid and deltamethrin decreased the biological productivity of *C. undecimnotata*. Zhang et al. reported the same biological pattern in *Bradysia odoriphaga* (Diptera: Sciaridae) with thiamethoxam [63]. This may be due to the pesticide affecting the accumulation of nutrients and/or growth of ovaries. Moreover, Devine et al. reported that a low dose of imidacloprid reduced aphid fecundity by inhibiting the production and viability of the nymphs [64].

Compatibility in terms of a pest’s natural enemies and pesticides is one of the major concerns of IPM strategies, and information about the effect of pesticides on the targeted pests and the non-target organism is a necessity [12]. Our results indicate that imidacloprid, and especially deltamethrin, applied at the recommended doses to a crop, may be harmful to *C. undecimnotata*. Our information could lead to a stronger understanding of the population prevalence patterns of *C. undecimnotata* treated with imidacloprid or deltamethrin in the field.

## 5. Conclusions

The present study demonstrated that imidacloprid and deltamethrin are toxic to *C. undecimnotata* because they increased pre-adult developmental time and reduced their population growth parameters. The determined toxicity of both insecticides tested (LD_10_ and LD_30_) was lower than the minimum recommended dosage for cover sprays. Therefore, the appropriate imidacloprid and deltamethrin field dosages should be carefully calculated for the conservation of the predator *C. undecimnotata*. However, the effect on field population levels remains unclear. Ongoing studies need to evaluate whether sublethal effects and/or hormesis of imidacloprid and deltamethrin affect later generations of *C. undecimnotata* and the aphid pest. Moreover, the effect of different methods of exposure to insecticides needs to be evaluated.

## Figures and Tables

**Figure 1 insects-12-00696-f001:**
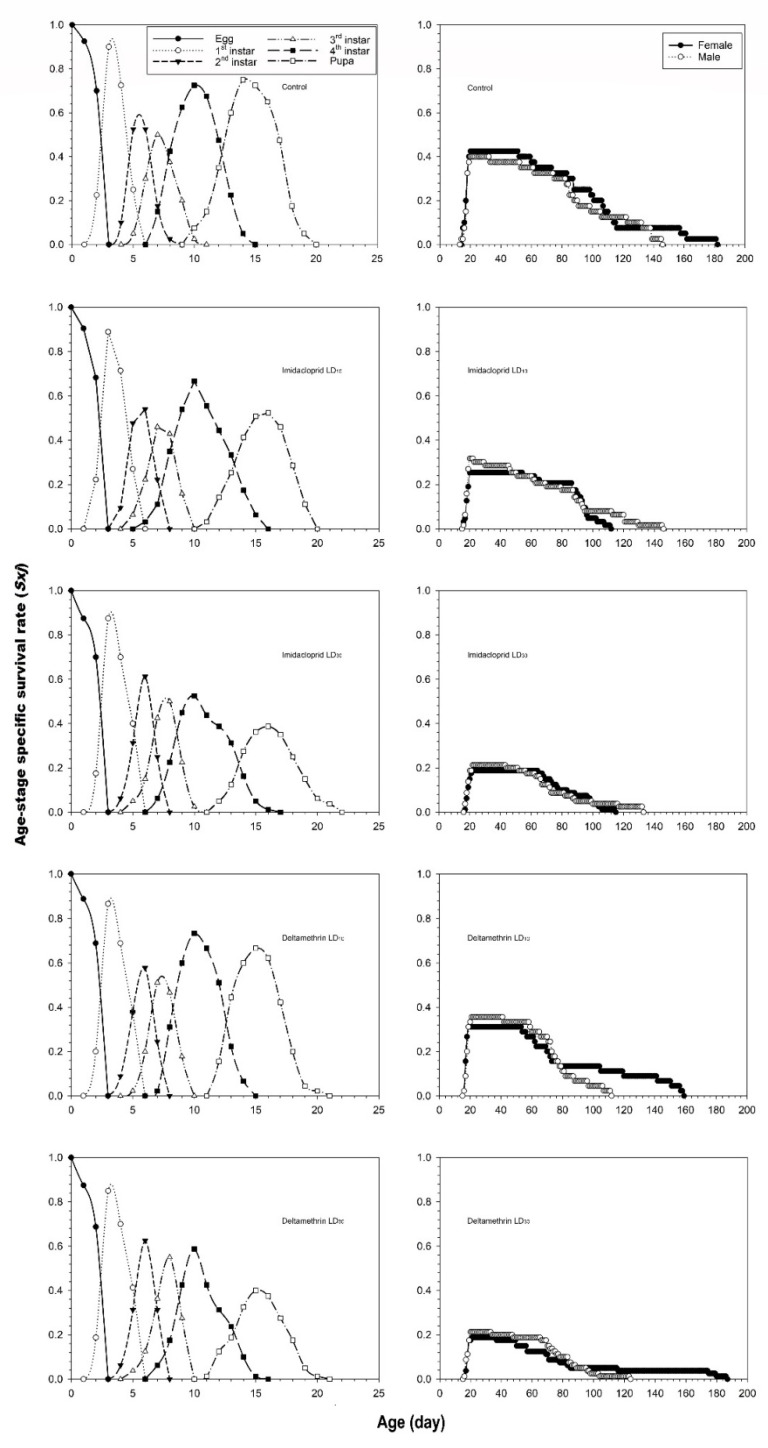
Age-stage specific survival rate (*s_xj_*) after fourth instar *C. undecimnotata* larvae exposed to sublethal and low imidacloprid and deltamethrin doses.

**Figure 2 insects-12-00696-f002:**
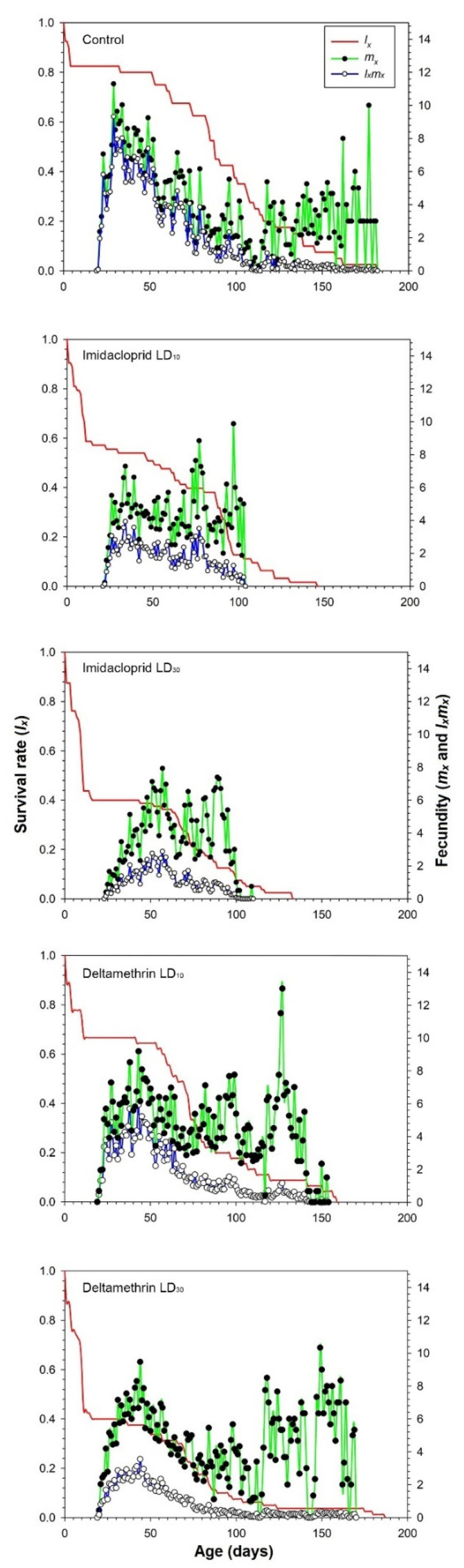
Age-specific survival rate (*lx*), age-specific fecundity *(m_x_*), and age-specific maternity (*l_x_m_x_*) after fourth instar *C. undecimnotata* larvae exposed to sublethal and low imidacloprid and deltamethrin doses.

**Figure 3 insects-12-00696-f003:**
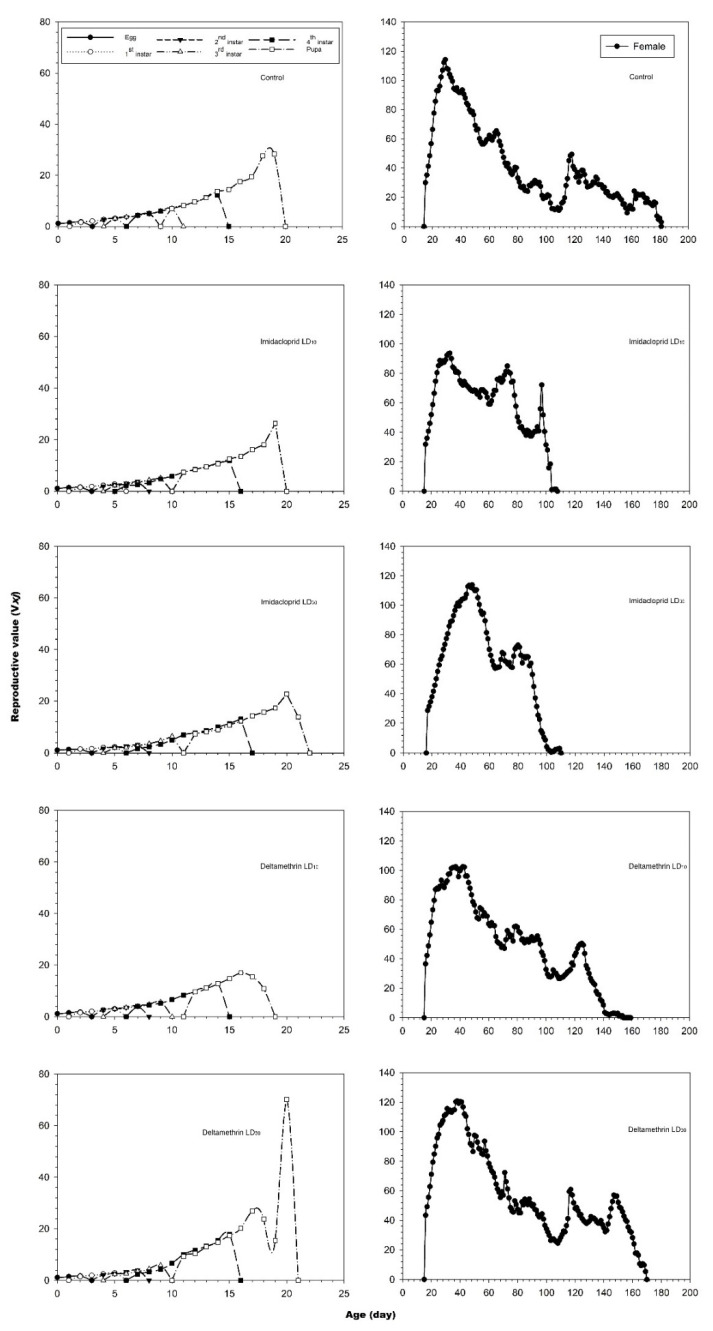
Age-stage specific reproductive values (*V_xj_*) values after fourth instar *C. undecimnotata* larvae exposed to sublethal and low imidacloprid and deltamethrin doses.

**Figure 4 insects-12-00696-f004:**
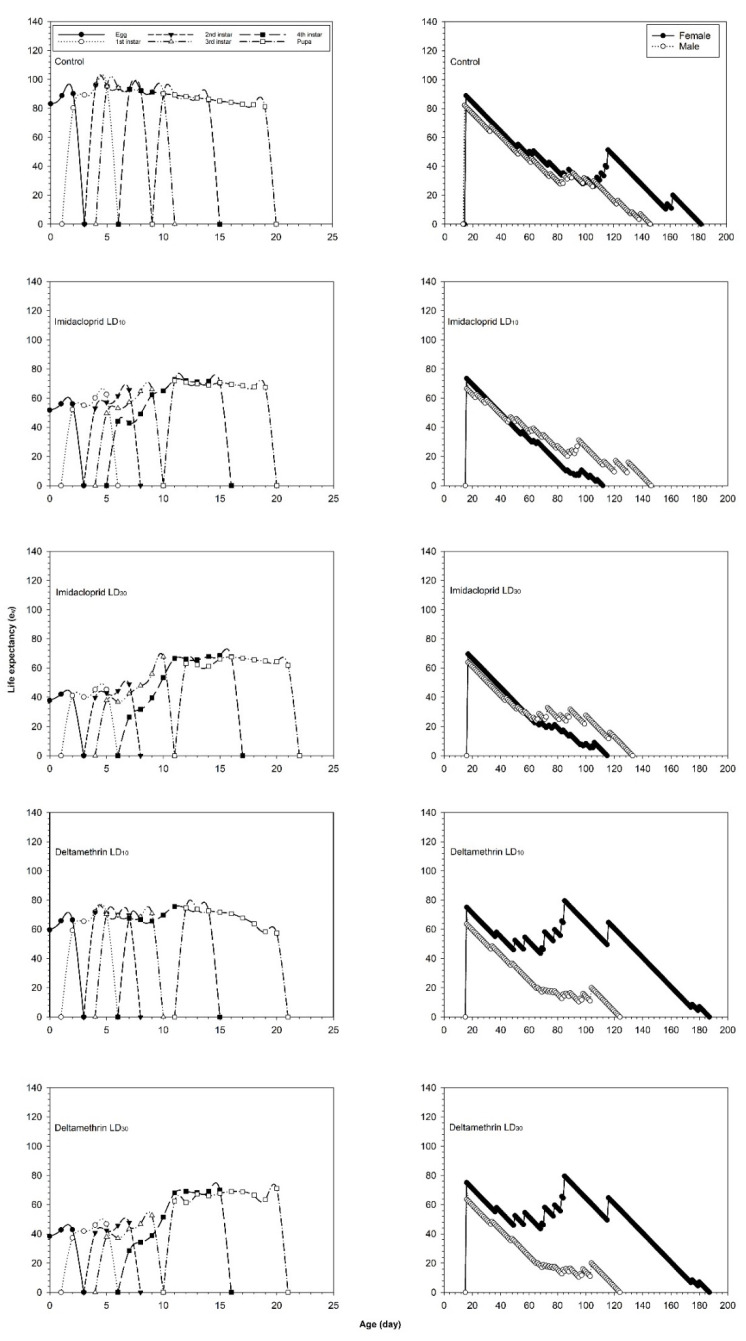
Life expectancy (e_xj_) values after fourth instar *C. undecimnotata* larvae exposed to sublethal and low imidacloprid and deltamethrin doses.

**Figure 5 insects-12-00696-f005:**
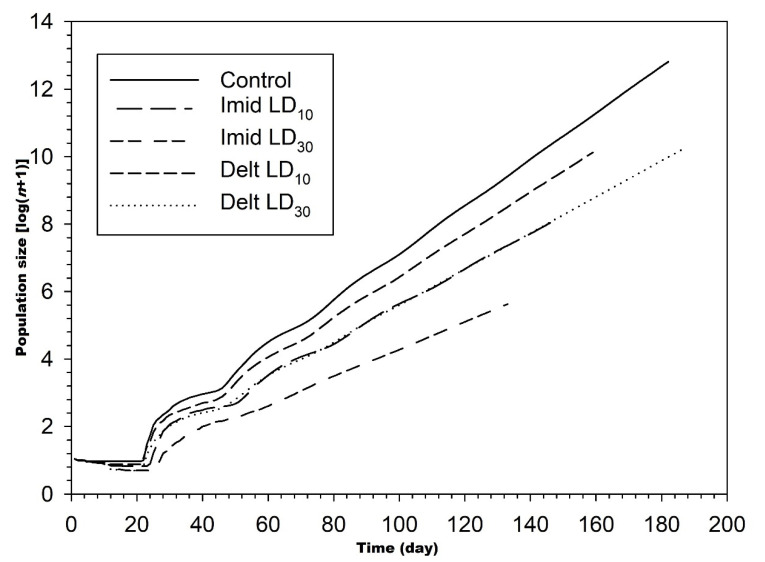
Population projection for *C. undecimnotata* larvae exposed to sublethal and low imidacloprid and deltamethrin doses.

**Table 1 insects-12-00696-t001:** Response (LD_10_, LD_30_, LD_50_) of fourth instars larval of *Ceratomegilla*
*undecimnotata* to different doses (dose ng a.i./insect) of imidacloprid and deltamethrin in lab bioassays.

Insecticide	N ^a^	Dose ng a.i. Insect ^−1^ (95% Confidence Limits)^−1^			Slope ± SE	χ2	*p*	df
		LD_10_	LD_30_	LD_50_				
Imidacloprid	320	10.68 (7.60–13.74)	20.95 (16.70–25.34)	33.41 (27.75–40.21)	2.588 ± 0.256	1.164	0.948	5
Deltamethrin	400	0.77 (0.54–1.02)	1.76 (1.38–2.16)	3.10 (2.53–3.81)	2.128 ± 0.189	2.913	0.893	7

^a^ Number of insects tested, SE: standard error, df: degrees of freedom.

**Table 2 insects-12-00696-t002:** Sublethal and low dose effects of imidacloprid and deltamethrin on the developmental time of fourth instar larva (days) and pupa (days) of *Ceratomegilla*
*undecimnotata* adults exposed to the insecticides from the fourth instar larval stage. Data are mean values ± SE.

Treatments	N	Development Time of Fourth Instar Larva (Days)	N	Development Time of Pupa (Days)
Control	33	4.06 ± 0.12 ^c^	33	4.85 ± 0.09 ^ab^
Imidacloprid LD_10_	37	5.22 ± 0.21 ^a^	36	4.75 ± 0.15 ^ab^
Imidacloprid LD_30_	35	5.31 ± 0.13 ^a^	32	5.03 ± 0.11 ^a^
Deltamethrin LD_10_	30	4.53 ± 0.13 ^b^	30	4.77 ± 0.12 ^ab^
Deltamethrin LD_30_	35	4.66 ± 0.14 ^b^	32	4.75 ± 0.08 ^b^

Means followed by the same letter in the same column are not significantly different based on the paired bootstrap test at the 5% significance level (*p* < 0.05). The sample size (N) is the number of individuals included in the calculation of the respective statistics.

**Table 3 insects-12-00696-t003:** Sublethal and low dose effects of imidacloprid and deltamethrin on the life parameters of fecundity (eggs/female), APOP (days), TROP (days), female adult longevity (days), and male adult longevity (days) of *Ceratomegilla*
*undecimnotata* adults exposed to the insecticides from the fourth instar larval stage. Data are mean values ± SE.

Treatments	N ^a^	Fecundity (Eggs/Female)	APOP (Days)	TPOP (Days)	Female Adult Longevity (Days)	N ^b^	Male Adult Longevity (Days)
Control	17	773.24 ± 82.15 ^a^	4.76 ± 0.28 ^bc^	22.18 ± 0.40 ^c^	86.53 ± 8.86 ^a^	16	78.69 ± 8.17 ^a^
Imidacloprid LD_10_	16	591.13 ± 63.80 ^ab^	5.88 ± 0.54 ^b^	24.31 ± 0.45 ^b^	71.06 ± 3.88 ^ab^	20	64.10 ± 7.49 ^ab^
Imidacloprid LD_30_	15	476.60 ± 87.41 ^b^	9.00 ± 1.01 ^a^	28.27 ± 1.07 ^a^	67.27 ± 4.19 ^b^	17	61.88 ± 6.40 ^ab^
Deltamethrin LD_10_	14	723.79 ± 132.05 ^ab^	4.43± 0.37 ^c^	22.00 ± 0.47 ^c^	79.79 ± 10.81 ^ab^	16	59.06 ± 4.41 ^b^
Deltamethrin LD_30_	15	674.20± 131.06 ^ab^	4.47± 0.34 ^c^	22.73 ± 0.40 ^c^	72.08 ± 12.92 ^ab^	17	61.41 ± 4.98 ^ab^

Means followed by the same letters in the same column are not significantly different based on the paired bootstrap test at the 5% significance level (*p* < 0.05). The sample size (N) is the number of individuals included in the calculation of the respective statistics.

**Table 4 insects-12-00696-t004:** Sublethal and low dose effects of imidacloprid and deltamethrin on the population growth parameters of intrinsic rate of increase. day^−1^, net reproductive rate (R_0_), mean generation time (days) and finite rate of increase. day^−1^ of *Ceratomegilla*
*undecimnotata* adults exposed to the insecticides from the fourth instar larval stage. Data are mean values ±SE.

Treatments	N	Intrinsic Rate of Increase (*r*) (Day)^−1^	Net Reproductive Rate (*R_0_*) (Offspring/Individual)	Mean Generation Time (*T*) (Days)	Finite Rate of Increase (*λ*) (Day)^−1^
Control	40	0.1590 ± 0.0078 ^a^	328.63 ± 69.51 ^a^	36.43 ± 1.03 ^c^	1.1724 ± 0.0091 ^a^
Imidacloprid LD_10_	63	0.1229 ± 0.0079 ^b^	150.13 ± 36.07 ^bc^	40.78 ± 1.23 ^b^	1.1308 ± 0.0079 ^b^
Imidacloprid LD_30_	80	0.0934 ± 0.0075 ^c^	89.36 ± 26.13 ^c^	48.11 ± 1.98 ^a^	1.0979 ± 0.0082 ^c^
Deltamethrin LD_10_	45	0.1431 ± 0.010 ^ab^	225.18 ± 63.79 ^ab^	37.87 ± 1.35 ^bc^	1.1538 ± 0.0115 ^ab^
Deltamethrin LD_30_	80	0.1234 ± 0.0091 ^b^	126.41 ± 37.90 ^bc^	39.24 ± 1.85 ^bc^	1.1313 ± 0.0103 ^b^

Means followed by the same letters in the same column are not significantly different based on the paired bootstrap test at the 5% significance level (*p* < 0.05). The sample size (N) is the number of individuals included in the calculation of the respective statistics.

## Data Availability

Not applicable.

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
