# Peer review of "Toxicity, Sublethal and Low Dose Effects of Imidacloprid and Deltamethrin on the Aphidophagous Predator Ceratomegilla undecimnotata (Coleoptera: Coccinellidae)"

_insects, 2021, doi:10.3390/insects12080696_

Round 1
Reviewer 1 Report
The manuscript entitled "Toxicity, sublethal and low dose effects of imidacloprid and deltamethrin on the aphidophagous predator Ceratomegilla undecimnotata (Coleoptera: Coccinellidae) " describes the detrimental effects on life parameters of this predator. The idea is good and the toxicological analysis mentioned supports the sublethal effects. The manuscript has certain shortcomings that need to be incorporated.
A few points:
Ls.22-23: Insert one conclusive sentence.
L.27: Which sublethal effects? Explain.
Ls.34-35: …negative influence? Which?
Ls.37-38: Keywords should be in alphabetic order. Also, keywords serve to widen the opportunity to be retrieved from a database. To put words that already are into title and abstracts makes KW not useful. Please choose terms that are neither in the title nor in abstract.
Ls.48-56: A brief information about deltamethrin is needed.
L.103: Delete “approximately”
L.109:… (30 cm diameter × 50 cm height)…
L.112, 122: Again, change “x” by “×”
L.140: …from the lethal toxicity bioassay.
L.180: Both techniques were…
Ls.216-222: … extended significantly? was significantly higher? Please, rephrase this paragraph.
L.306: …larvae were half…
L.311: Deltamethrin was the…
L.324, 347: Repetitive, delete “significantly”
L:379: …that imidacloprid and deltamethrin are toxic…
Figures 1, 2, 3, and 4 should be organized in one only plate.
Reviewer 2 Report
The paper “Toxicity, sublethal and low dose effects of imidacloprid and deltamethrin on the aphidophagous predator Ceratomegilla undecimnotata (Coleoptera: Coccinellidae)” is a well-designed comprehensive study about the lethal and sublethal effects of two insecticides, whose use was recently strongly limited in EU. Introduction and literature revision are adequate. The experimental set up and data analysis seem appropriate, and results are sufficiently discussed. Overall, I believe this paper could be suitable for publication on Insects after revision.
- Simple summary and Abstract are very similar. I believe the SS should a bit revised to be more consistent with its scope (give a brief comprehensive idea to readers of the context and importance of this research).
- Subsections in the section 2. Material and Methods should be numbered (e.g., 2.1, 2.1.1.).
- LN 152 “Egg hatchability” here is registered for before treatment F1 eggs? This parameter has been considered also for F2 eggs?
- LN 187-8 These data are reported in Table 1; I believe they are not needed here.
- LN 191-3 It is unclear to me the statistical values reported here what are referring to. Were there no statistical differences between what?
- Result section: the statistical values are needed to support all this section (please, see the attached file). I can see, from the tables, letters from paired bootstrap test, but neither a P value was provided... When a parameter is reported to SIGNIFICANTLY or NOT SIGNIFICANTLY differ from another the statistic references is always necessary.
- Minor comments are listed in the attached pdf file.

Reviewer 3 Report
The manuscript « Toxicity, sublethal and low dose effects of imidacloprid and deltamethrin on the aphidophagous predator Ceratomegilla undecimnotata (Coleotera: Coccinellidae) » by Skouras and coworkers describes a study where they analyze the effect of two insecticides into the survival, development, reproduction, longevity and population parameters of the natural enemy Ceratomegilla undecimnotata.
Major corrections:
- Even the experiments are complete and well done and the results are very interesting, the Results section is missing some information coming from Tables and Figures. Here you are some examples:
The Imidacloprid LD10 effect in TPOP. Data from Table 4 is also missing in the 3.3 section, such as RO or T. In line 244 you mention imidacloprid but no deltamethrim. I don’t understand the correspondence between the days that are mentioned in the Sxj paragraph for the control (32/13 days) vs. the Figure 1 graph, where the adults go to 140/180 days. Lxmx is not commented from Figure 2. The effect of deltamethrim in males and females exj is not mentioned.
- Moreover, the Discussion section has to be rewritten almost completely, because there are some incorrect sentences and others that are difficult to understand. Here you are some examples:
Lines 310-314 “Deltamethrin were the most toxic and the same toxic effect were observed in other insecticides that had the same mode of action in many coccinellids such as A. bipunctata.”
Lines 328-329: reference?
Line 337: developmental time is prolonged, not reduced.
Lines 342-343: why you take that conclusion?
Lines 352-354 “Imidacloprid extended the APOP and TPOP in LD10 and LD30 treatment and reduced the intrinsic rate of increase (r) this was the lowest, when compared LD10 and LD30 with the control treatments.
Lines 360-361. “the sxj value was also affected; this measure of survival rate decreased in accordance with the increase in imidacloprid and dose in fourth instar larvae, pupae and adults.”
Line 374: why specially deltamethrim, if imidacloprid has more effects?
Minor corrections:
Line 34: “have” instead of “has”
Lines 54-56: Rewrite please. Moreover, deltamethrin is not mentioned in the Intro.
Line 83: age, stage and sex
Line 84-85: Rewrite please. Moreover, “do not take” instead of “doesn´t takes”
Line 99: “doses” instead of “dose”
Line 189: “a.i. per insect” instead of “imidacloprid per insect”
Table 1: Why the dose units are mg/liter instead of ng a.i./insect?
Line 205-207. Move to the beginning of the paragraph.
Line 203. A mark point is missing between “group” and “The”
Line 205: “imid or delta” intead of “imid and delta”
Line 205: “groups” instead of “group”
Table 3: I would put data in the table in the same order as you mention in the text.
Line 232: Add “and deltamethrin” in the title.
Line 379: “are” instead of “is”
Line 383: especially what? There is a gap here.
